# Measuring and regularizing networks in function space

Ari S. Benjamin[*1], David Rolnick[1], and Konrad P. Kording[1]

[1]University of Pennsylvania, Philadelphia, PA, 19142

## Abstract

To optimize a neural network one often thinks of optimizing its parameters, but it is ultimately a matter of optimizing the function that maps inputs to outputs. Since a change in the parameters might serve as a poor proxy for the change in the function, it is of some concern that primacy is given to parameters but that the correspondence has not been tested. Here, we show that it is simple and computationally feasible to calculate distances between functions in a $L^2$ Hilbert space. We examine how typical networks behave in this space, and compare how parameter $\ell^2$ distances compare to function $L^2$ distances between various points of an optimization trajectory. We find that the two distances are nontrivially related. In particular, the $L^2/\ell^2$ ratio decreases throughout optimization, reaching a steady value around when test error plateaus. We then investigate how the $L^2$ distance could be applied directly to optimization. We first propose that in multitask learning, one can avoid catastrophic forgetting by directly limiting how much the input/output function changes between tasks. Secondly, we propose a new learning rule that constrains the distance a network can travel through $L^2$-space in any one update. This allows new examples to be learned in a way that minimally interferes with what has previously been learned. These applications demonstrate how one can measure and regularize function distances directly, without relying on parameters or local approximations like loss curvature.

## 1 Introduction

A neural network's parameters collectively encode a function that maps inputs to outputs. The goal of learning is to converge upon a good input/output function. In analysis, then, a researcher should ideally consider how a network's input/output function changes relative to the space of possible functions. However, since this space is not often considered tractable, most techniques and analyses consider the parameters of neural networks. Most regularization techniques, for example, act directly on the parameters (e.g. weight decay, or the implicit constraints stochastic gradient descent (SGD) places upon movement). These techniques are valuable to the extent that parameter space can be taken as a proxy for function space. Since the two might not always be easily related, and since we ultimately care most about the input/output function, it is important to develop metrics that are directly applicable in function space.

In this work we show that it is relatively straightforward to measure the distance between two networks in function space, at least if one chooses the right space. Here we examine $L^2$-space, which is a Hilbert space. Distance in $L^2$ space is simply the expected $\ell_2$ distance between the outputs of two functions when given the same inputs. This computation relies only on function inference.

Using this idea of function space, we first focus on characterizing how networks move in function space during optimization with SGD. Do random initializations track similar trajectories? What happens in the overfitting regime? We are particularly interested in the relationship between trajectories in function space and parameter space. If the two are tightly coupled, then parameter change can be taken as a proxy for function change. This common assumption (e.g. Lipschitz bounds) might not always be the case.

---

[*]aarrii@seas.upenn.edu

Next, we demonstrate two possibilities as to how a function space metric could assist optimization. In the first setting we consider multitask learning, and the phenomenon of catastrophic forgetting that makes it difficult. Many well-known methods prevent forgetting by regularizing how much the parameters are allowed to shift due to retraining (usually scaled by a precision matrix calculated on previous tasks). We show that one can instead directly regularize changes in the input/output function of early tasks. Though this requires a "working memory" of earlier examples, this scheme turns out to be quite data-efficient (and more so than actually retraining on examples from old tasks).

In the second setting we propose a learning rule for supervised learning that constrains how much a network's function can change any one update. This rule, which we call *Hilbert-constrained gradient descent* (HCGD), penalizes each step of SGD to reduce the magnitude of the resulting step in $L^2$-space. This learning rule thus changes the course of learning to track a shorter path in function space. If SGD generalizes in part because large changes to the function are prohibited, then this rule will have advantages over SGD. Interestingly, HCGD is conceptually related to the natural gradient. As we derive in §3.2.1, the natural gradient can be viewed as resulting from constrains changes in a function space measured by the Kullbeck-Leibler divergence.

## 2 EXAMINING NETWORKS IN FUNCTION SPACE

We propose to examine the trajectories of networks in the space of functions defined by the inner product $\langle f, g \rangle = \int_{\mathbb{X}} f(x)g(x)d\mu(x)$, which yields the following norm:

$$\|f\|^2 = \int_{\mathbb{X}} |f|^2 d\mu.$$

Here $\mu$ is a measure and corresponds to the probability density of the input distribution $\mathbb{X}$. Note that this norm is over an empirical distribution of data and not over the uniform distribution of all possible inputs. The $|\cdot|^2$ operator refers to the 2-norm and can apply to vector-valued functions. While we refer to this space as a Hilbert space, we make no use of an inner product and can also speak of this as any normed vector space, e.g. a Banach space. This norm leads to a notion of distance between two functions $f$ and $g$ given by

$$\|f - g\|^2 = \int_{\mathbb{X}} |f - g|^2 d\mu.$$

Since $\mu$ is a density, $\int_{\mathbb{X}} d\mu = 1$, and we can write

$$\|f - g\|^2 = \mathbb{E}_{\mathbb{X}}[|f(x) - g(x)|^2].$$

The expectation can be approximated as an empirical expectation over a batch of examples drawn from the input distribution:

$$\|f - g\|^2 \approx \frac{1}{N} \sum_{i=0}^{N} |f(x_i) - g(x_i)|^2.$$

The quality of the empirical distance, of course, will depend on the shape and variance of the distribution of data as well as the form of $f$ and $g$. In section 2.3, we empirically investigate the quality of this estimator for reasonably sample sizes $N$.

### 2.1 DIVERGENCE OF NETWORKS IN $L^2$-SPACE DURING TRAINING

We wish to compare at high level how networks move through parameter and function space. Our first approach is to compare a low-dimensional embedding of the trajectories through these spaces. In Figure 1, we take a convolutional neural network and train three random initializations on a 5000-image subset of CIFAR-10. By saving the parameters of the network at each epoch as well as the output on a single large validation batch, we can later compute the $\ell^2$ parameter distance and the

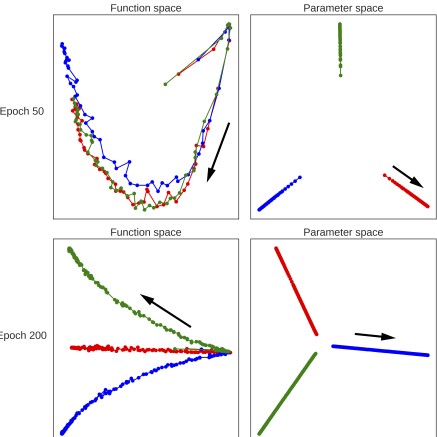

Figure 1:
Visualization of the trajectories of three random initializations of a network through function space, left, and parameter space, right. The network is a convolutional network trained on a 5,000 image subset of CIFAR-10. At each epoch, we compute the $L^2$ and $\ell^2$ distances between all previous epochs, forming two distance matrices, and then recompute the 2D embedding from these matrices using multidimensional scaling. Each point on the plots represents the network at a new epoch of training.The black arrows represent the direction of movement.

$L^2$ function distance between the snapshots of network at each epoch. The resulting distance matrix is then visualized as a two-dimensional embedding.

In parameter space, the networks are initialized at very different points and proceed to diverge yet further from these points. Despite this divergence, each trajectory yields a network that has learned the training data perfectly and generalizes with $\sim 50\%$ accuracy to a test set. This illustrates the wide range of parameter settings that can be used to represent a given neural network function. The behavior of the same initializations in function space is quite different. First, note that all three initializations begin at approximately the same point in function space. This is an intriguing property of random initializations that, rather than encoding entirely random functions, random sets of parameters lead on average to the same function (for related work, see e.g. Giryes et al. (2016)). The initializations then largely follow an identical path for the initial stage of learning. Different initializations thus learn in similar manners, even if the distance between their parameters diverges. During late-stage optimization, random initializations turn from a shared trajectory and begin to diverge in $L^2$ space. These differences underlie the general principle that $L^2$ distances behave differently than $\ell^2$ distances, and that functional regularization could assist training and reduce overfitting.

## 2.2   COMPARING $L^2$ FUNCTION DISTANCE WITH $\ell^2$ PARAMETER DISTANCE

How well do parameter distances reflect function distances? The answer to this question is relevant for any method that directly considers the behavior of parameters. Certain theoretical analyses, furthermore, desire bounds on function distances but instead find bounds on parameter distances and relate the two with a Lipschitz constant (e.g. Hardt et al. (2015)). Thus, for theoretical analyses and optimization methods alike, it is important to empirically evaluate how well parameter distances correspond to function distances in typical situations.

We can compare these two situations by plotting a change in parameters $\|\Delta\theta\|$ against the corresponding change in the function $\|f_\theta - f_{\theta+\Delta\theta}\|$. In Figure 2 we display this relation for several relevant distance during the optimization of a CNN on CIFAR-10. There are three scales: the distance between individual updates, the distance between epochs, and the distance from initialization.

Note, first, that networks continue to move in function space as well as in parameter space after test error converges, which is around epoch 60. (The test error can be seen in Appendix A, along with identical plots colored by test error instead of epoch.) Their movement relative to initialization slows, but there is still large movement relative to previous iterations and previous epochs.

What changes strikingly throughout optimization is the relationship between parameter and function space. There is a qualitative difference in the ratio of parameter distances to function distances that visible at all three distance scales. Early epochs generally see larger changes in $L^2$ space for a given change in parameters. Intriguingly, the ratio of the two distances appears to converge to a single value at late optimization, after test error saturates. This is not because the network ceases to move, as noted above. Rather, the loss landscape shifts such that this ratio become constant.

It is also clear from these plots that there is not a consistent positive correlation between the parameter and function distances between any two points on the optimization trajectory. For example, the parameter distance between successive epochs is negatively correlated with the $L^2$ distance for most of optimization (Fig. 2b). The distance from initialization shows a clean and positive relationship, but the relationship changes during optimization. Between successive batches, $L^2$ distance correlates with parameter distance at late epochs, but less so early in optimization when learning is quickest. Thus, at different stages of optimization, the $L^2/\ell^2$ ratio is often quite different.

The usage of Batch Normalization (BN) and weight decay in this analysis somewhat affects the trends in the $L^2/\ell^2$ ratio. In Appendix A we reproduce these plots for networks trained without BN and without weight decay. The overall message that the $L^2/\ell^2$ ratio changes during optimization is unchanged. However, these methods both change the scale of updates, and appear to do so differently throughout optimization, and thus some trends are different. In Appendix B, we also isolate the effect of training data, by reproducing these plots for a CNN trained on MNIST and find similar trends. Overall, the correspondence between parameter and function distances depends strongly on the context.

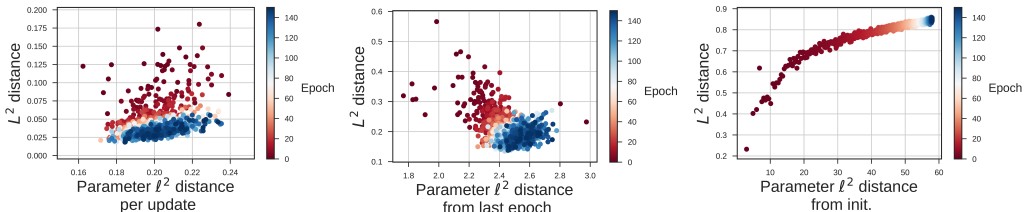

Figure 2: Parameter distances is sometimes, but not always, representative of function distances. Here we compare the two at three scales during the optimization of a CNN on CIFAR-10. Left: Distances between the individual SGD updates. Middle: Distances between each epoch. Right: Distances from initialization. On all three plots, note the changing relationship between function and parameter distances throughout optimization. The network is the same as in Figure 1: a CNN with four convolutional layers with batch normalization, followed by two fully-connected layers, trained with SGD with learning rate = 0.1, momentum = 0.9, and weight decay = 1e-4. Note that the $L^2$ distance is computed from the output after the softmax layer, meaning possible values range from 0 to 1.

### 2.3 CONVERGENCE OF THE EMPIRICAL ESTIMATOR

It might be worried that since function space is of infinite dimension, one would require prohibitively many examples to estimate a function distance. However, we find that one can compute a distance between two functions with a relatively small amount of examples. Figure 3 shows how the estimated $L^2$ distance converges with an increasing number examples. In general, we find that only a few hundred examples are necessary to converge to an estimation within a few percent.

## 3 APPLICATIONS

### 3.1 COMBATTING CATASTROPHIC FORGETTING IN AN ONLINE LEARNING TASK (WITH WORKING MEMORY)

If, after having been trained on a task, a neural network is retrained on a new task, it often forgets the first task. This phenomenon is termed 'catastrophic forgetting'. It is the central difficulty of multitask training as well as applications requiring that learning be done online (especially in non-IID situations). Essentially, new information must be encoded in the network, but the the information pertinent to the previous task must not be overwritten.

Most efforts to combat catastrophic forgetting rely on restricting how much parameters can change between tasks. Elastic Weight Consolidation (EWC; Kirkpatrick et al. (2017)), for example, adds a penalty to the loss on a new task $B$ that is the distance from the weights after learning on an earlier

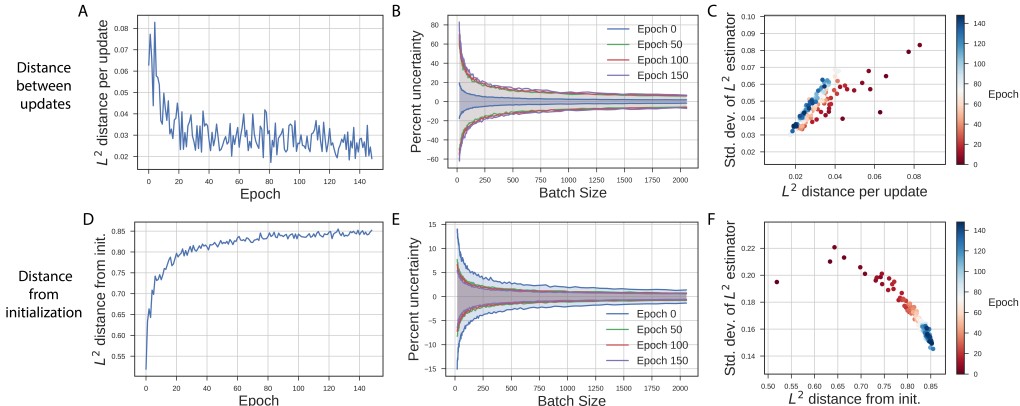

Figure 3: The variance of the the $L^2$ estimator is small enough that it can be reasonably estimated from a few hundred examples. In panels A and D, we reproduced $L^2$ distances seen in the panels of Fig. 2. As we increase the number of validation examples these distances are computed over, the estimations become more accurate. Panels B and E show the 95% confidence bounds for the estimation; on 95% of batches, the value will lie bewteen these bounds. These bounds can be obtained from the standard deviation of the $L^2$ distance on single examples. In panel C we show that the standard deviation scales linearly with the $L^2$ distance when measured between updates, meaning that a fixed batch size will often give similar percentage errors. This is not true for the distance from initialization, in panel F; early optimization has higher variance relative to magnitude, meaning that more examples are needed for the same uncertainty. In the Appendix, we also display the convergence of the $L^2$ distance estimator between epochs.

task A, multiplied by the diagonal of the Fisher information matrix $F$ (calculated on task $A$):

$$L_{EWC}(\theta) = L_B(\theta) + \frac{\lambda}{2} \sum_i F_i(\theta_i - \theta_{i,A})^2$$

This idea is closely related to well-studied approaches to Bayesian online learning, if $F$ is interpreted as a precision matrix (Honkela & Valpola (2003), Opper & Winther (1998)). Other similar approaches include that of Ritter et al. (2018), who use a more accurate approximation of the Fisher, and Synaptic Intelligence (SI; Zenke et al. (2017)), which discounts parameter change via a diagonal matrix in which each entry reflects the sum contribution of that parameter to the loss. Each of these method discourages catastrophic forgetting by restricting movement in parameter space between tasks, scaled by a (perhaps diagonal) precision matrix calculated on previous tasks.

Using a function space metric, it is not hard to ensure that the network's output function on previous tasks does not change during learning. In this case, the loss for a new task $B$ is modified to be:

$$L(\theta) = L_B(\theta) + \frac{\lambda}{2} \|f_{\theta_A} - f_{\theta_B}\|$$

The regularization term is the $L^2$ distance between the current function $f_{\theta_B}$ and the function after training on task A, $f_{\theta_A}$. Since our function space metric is defined over a domain of examples, we will store a small set of previously seen examples in a working memory, as well as the output on those examples. This memory set will be used to calculate the $L^2$ distance between the current iteration and the snapshot after training. This is a simple scheme, but novel, and we are not aware of direct precedence in the literature.

A working memory approach is employed in related work (Lopez-Paz et al. (2017); Rebuffi et al. (2017)). Note, however, that storing old examples violates the rules of strict online learning. Nevertheless, for large networks it will be more memory-efficient. EWC, for example, requires storing a snapshot of each parameter at the end of the previous task, as well as a diagonal precision matrix with as many entries as parameters. For the 2 hidden layer network with 400 nodes each that was used in the MNIST task in Kirkpatrick et al. (2017), this is 1,148,820 new parameters, or as many pixels as 1,465 MNIST images. When each layer has as many as 2,000 nodes, as in Fig. 3B of Kirkpatrick

et al. (2017), the extra stored parameters are comparable to 15,489 MNIST images. The working memory approach that is required to regularize function change from old tasks is thus comparable or cheaper in memory.

### 3.1.1 EMPIRICAL RESULTS

We compared the performance of our approach at the benchmark task of permuted MNIST. This task requires a single MLP to learn to classify a sequence of MNIST tasks in which the pixels have been randomly permuted differently on each task. We trained an MLP with 2 hidden layers, 400 nodes each, for 10 epochs on each of 8 such permuted datasets. In Figure 4, we display how the test accuracy on the first of 8 tasks degrades with subsequent learning.

To build the working memory, we keep 1024 examples from previous tasks, making sure that the number of examples from each task is equal. We also remember the predictions on those examples at the end of training on their originating tasks. To calculate the $L^2$ distance, we simply re-infer on the examples in working memory, and regularize the distance from the current outputs to the remembered outputs. We chose $\lambda = 1.3$ as the regularizing hyperparameter from a logarithmic grid search.

In Figure 4, we compare this method to four comparison methods. The "ADAM" method is ADAM with a learning rate of 0.001, which nearly forgets the first task completely at the end of the 8 tasks. The "ADAM+retrain" method is augmented with a working memory of 1024 examples that are stored from previous tasks. Every $n$ iterations (we found $n = 10$ to be best), a step is taken to decrease the loss on the memory cache. This method serves as a control for the working memory concept. We also include EWC and SI as comparisons, using the hyperparameters used in their publications ($\lambda = 500, \epsilon = c = 0.1$). Overall, we found that regularizing the $L^2$ distance on a working memory cache was more successful than simply retraining on the same cache. It also outperformed EWC, but not SI. Note that these methods store diagonal matrices and the old parameters, and in this circumstance these were larger in memory than the memory cache.

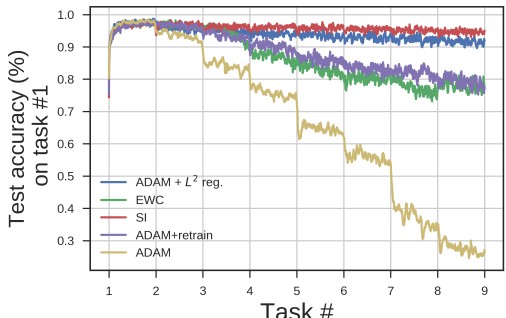

Figure 4: Regularizing the $L^2$ distance from old tasks (calculated over a working memory cache of size 1024) can successfully prevent catastrophic forgetting. Here we display the test performance on the first task as 7 subsequent tasks are learned. Our method outperforms simply retraining on the same cache (ADAM+retrain), which potentially overfits to the cache. Also displayed are ADAM without modifications, EWC, and SI.

### 3.2 CONSTRAINING CHANGES IN $L^2$ DURING LEARNING

In this section we propose that the $L^2$ distance can be used for regularization in a single supervised task. In the space of parameters, SGD is a strongly local update rule and large jumps are generally prohibited. SGD is thus more likely to find solutions that are close to the initialization, and furthermore to trace a path of limited length. This discourages the sampling a large volume of parameter space during optimization. If the mapping between parameter and function space is not already very tight, and locality is important for generalization, then additionally constricting changes in function space should help.

On the basis of this logic, we propose a learning rule that directly constrains the path length of the optimization trajectory $L^2$ space. If a network would have been trained to adjust the parameters $\theta$ to minimize some cost $C_0$, we will instead minimize at each step $t$ a new cost given by:

$$C = C_0 + \lambda \| f_{\theta_t} - f_{\theta_t + \Delta\theta} \| \tag{1}$$

Like all regularization terms, this can also be viewed as a Langrangian that satisfies a constraint. Here, this constraint ensures that the change in $L^2$-space does not exceed some constant value. To evaluate Equation 1, we can approximate the norm with an empirical expectation over $\mathbb{X}$:

$$C = C_0 + \lambda \left( \frac{1}{N} \sum_{i=0}^{N} |f_{\theta_t}(x_i) - f_{\theta_t + \Delta\theta}(x_i)|^2 \right)^{1/2}.$$

This cost function imposes a penalty upon the difference between the output of the current network at time $t$ and the proposed network at $t + 1$. The data $x_i$ may derive from some validation batch but must pull from the same distribution $\mathbb{X}$. It would also be possible to use unlabeled data.

We can write an update rule to minimize Equation 1 that is a modification of gradient descent. We call the rule Hilbert-constrained gradient descent (HCGD). It minimizes C in Equation 1 via an inner loop of gradient descent. To optimize $C$ via gradient descent, we first replace $C_0$ with its first order approximation $J^T \Delta\theta$, where $J$ is the Jacobian. Thus we seek to converge to a $\Delta\theta'$ at each update step, where

$$\Delta\theta' = \operatorname*{argmin}_{\Delta\theta} \left( J^T \Delta\theta + \frac{\lambda}{N} \sum_{i=0}^{N} |f_{\theta_t}(x_i) - f_{\theta_t + \Delta\theta}(x_i)|^2 \right) \qquad (2)$$

Minimization of the proper $\Delta\theta$ can be performed in an inner loop by a first order method. We first propose some $\Delta\theta_0 = -\epsilon J = -\epsilon \nabla_\theta C_0$ (for learning rate $\epsilon$) and then iteratively correct this proposal by gradient descent towards $\Delta\theta'$. If only one correction is performed, we simply add the derivative of the Hilbert-constraining term after $\Delta\theta_0$ has been proposed. We found empirically that a single correction was often sufficient. In Appendix C, we demonstrate that this algorithm does actually decrease the distance traveled in function space, as expected. This algorithm is shown in Algorithm 1.

---

Algorithm 1: Hilbert-constrained gradient descent. Implements Equation 2.

**Require:** $\epsilon$                  ▷ Overall learning rate
**Require:** $\eta$             ▷ Learning rate for corrective step
 1: **procedure**
 2:      $\theta \leftarrow \theta_0$                  ▷ Initialize parameters
 3:      **while** $\theta_t$ not converged **do**
 4:          draw $X \sim \mathbb{P}_x$          ▷ Draw training batch
 5:          $J \leftarrow \nabla_\theta C_0(X)$          ▷ Calculate gradients
 6:          $\Delta\theta_0 \leftarrow -\epsilon J$          ▷ Proposed update

 7:          draw $X_V \sim \mathbb{P}_x$          ▷ Draw validation batch
 8:          $g_{L^2} \leftarrow \nabla_{\Delta\theta} \left( \frac{\lambda^2}{N} \sum_{x_i \in X_V}^{N} |f_{\theta_t}(x_i) - f_{\theta_t + \Delta\theta}(x_i)|^2 \right)^{1/2}$      ▷ Calculate correction
 9:          $\Delta\theta' \leftarrow \Delta\theta_0 - \eta(g_{L^2})$
10:          $\theta_t \leftarrow \theta_{t-1} + \Delta\theta'$
11:      **return** $\theta_t$

---

Note that the "proposed update" is presented as an SGD step, but could be a step of another optimizer (e.g. ADAM). In the Appendix, we display an extended version of this algorithm. This version allows for multiple corrective iterations in each step. It also allows for a form of momentum. In standard momentum for SGD, one follows a "velocity" term $v$ which is adjusted at each step with the rule $v \leftarrow \beta v + \epsilon J$ (e.g. see Sutskever et al. (2013)). For HCGD, we also keep a velocity term but update it with the final Hilbert-constrained update $\Delta\theta$ rather than $\epsilon J$. The velocity is used to propose the initial $\Delta\theta_0$ in the next update step. We found that this modification of momentum both quickened optimization and lowered generalization error.

### 3.2.1 RELATION TO THE NATURAL GRADIENT

The natural gradient turns out to carry a similar interpretation as HCGD, in that the natural gradient also regularizes the change in functions' output distributions. Specifically, the natural gradient can be derived from a penalty upon the change in a network's output distribution as measured by the Kullbeck-Leibler divergence (rather than the $L^2$ distance).

To show this, we start with a similar goal of function regularization and will come upon the natural gradient. Let us seek to regularize the change in a network's output distribution $\mathbb{P}_\theta$ throughout optimization of the parameters $\theta$, choosing the Kullbeck-Leibler (KL) divergence as a measure of similarity between any two distributions. To ensure the output distribution changes little throughout optimization, we define a new cost function

$$C = C_0 + \lambda D_{KL}(\mathbb{P}_{\theta_{t+1}}\|\mathbb{P}_{\theta_t}) \tag{3}$$

where $C_0$ is the original cost function and $\lambda$ is a hyperparameter that controls the importance of this regularization term. Optimization would be performed with respect to the proposed update $\theta_{t+1}$.

Evaluating the KL divergence directly is problematic because it is infeasible to define the output density $\mathbb{P}_\theta$ everywhere. One can obtain a more calculable form by expanding $D_{KL}(\mathbb{P}_{\theta_{t+1}}\|\mathbb{P}_{\theta_t})$ around $\theta_t$ to second order with respect to $\theta$. The Hessian of the KL divergence is the Fisher information metric $F$. With $\Delta\theta \equiv (\theta_{t+1} - \theta_t)$, we can rewrite our regularized cost function as

$$C \approx C_0 + \frac{\lambda}{2}\Delta\theta^T F \Delta\theta \tag{4}$$

To optimize $C$ via gradient descent we first replace $C_0$ with its first order approximation.

$$C \approx J^T \Delta\theta + \frac{\lambda}{2}\Delta\theta^T F \Delta\theta \tag{5}$$

At each evaluation, $J$ is evaluated before any step is made, and we seek the value of $\Delta\theta$ that minimizes Equation 5. By setting the derivative with respect to $\Delta\theta$ to be zero, we can see that this value is

$$\Delta\theta = \frac{1}{\lambda}F^{-1}J \tag{6}$$

When $\lambda = 1$ this update is equal to the natural gradient. Thus, the natural gradient emerges as the optimal update when one regularizes the change in the output distribution during learning.

In Appendix E, we show how one can approximate the natural gradient with an inner first-order optimization loop, like in HCGD. We note that HCGD is computationally cheaper than the exact natural gradient. It does not require any matrix inversions, nor the calculation of separate per-example gradients. When the validation batch $X_V$ is drawn anew for each of $n$ corrective iterations (step 8 in Algorithm 1), HCGD requires an additional two forward passes and one backwards pass for each correction, for a total of $2 + 3n$ passes each outer step.

### 3.2.2 THE NATURAL GRADIENT IN THE LITERATURE

In addition to being seen as a regularizer of functional change, it in an interesting aside to note that variants of the natural gradient have appeared with many justifications. These include data efficiency, minimizing a regret bound during learning, speeding optimization, and the benefits of whitened gradients.

Amari originally developed the natural gradient in the light of information geometry and efficiency (Amari et al. (1996); Amari (1998)). If some directions in parameter space are more informative of the network's outputs than others, then updates should be scaled by each dimension's informativeness. Equivalently, if not all examples carry equal information about a distribution, then the update step should be modified to make use of highly informative examples. That is, we wish to find a Fisher-efficient algorithm (see Amari et al. (2000)). The natural gradient uses the Fisher information matrix to scale the update by parameters' informativeness.

There is also a connection between the natural gradient (and thus HCGD) and techniques that normalize and whiten gradients. The term $F^{-1}J$, after all, simply ensures that steps are made in a parameter space that is whitened by the covariance of the gradients. Whitening the gradients thus has the effect that SGD becomes more similar to the natural gradient. It appears that many approaches to normalize and whiten activations or gradients have been forwarded in the literature (Raiko et al. (2012);Simard et al. (1998); Schraudolph & Sejnowski (1996); Crammer et al. (2009); Wang et al. (2013); LeCun et al. (1991); Schraudolph (1998); Salimans & Kingma (2016)). A similar effect is able to be learned with Batch Normalization, as well (Ioffe & Szegedy (2015)). By normalizing and whitening the gradients, or by proxy, the activations, these various methods ensure that parameter space is a better proxy for function space.

### 3.3 EMPIRICAL COMPARISON OF HCGD

We compared HCGD and SGD on feedforward and recurrent architectures. If it is important that SGD limits changes in function space, and parameter and function space are loosely coupled, then HCGD should improve upon SGD. In all tests, we used a tuned learning rate $\epsilon$ for SGD, and then used the same learning rate for HCGD. We use values of $\lambda = 0.5$ and $\eta = 0.02$, generally about 10 times less than the principal learning rate $\epsilon$. (For the $n = 1$ version, $\lambda$ can be folded into the inner learning rate $\eta$. Values were chosen so that $\lambda\eta = 0.01$.) We chose the batch size for the "validation" batch to be 256. While the examples in each "validation" batch were different than the training batch, they were also drawn from the train set. All models were implemented in PyTorch (Paszke et al. (2017)).

We tested HCGD as applied to the CIFAR-10 image classification problem. For reproducibility, we trained a Squeezenet v1.1, a convolutional neural network model with batch normalization optimized for parameter efficiency (Iandola et al. (2016)). Overall HCGD does not outperform SGD in the final learning stage when trained with the same learning rate as SGD (initial $\epsilon = 0.1$), though it does perform better in the early stage while the learning rate is high (Figure 5). When we increase the initial learning rate to $\epsilon = 0.3$ (red trace), the training accuracy decreases but the test accuracy is still marginally higher than SGD. Given the difference in relative performance between the high and low learning rate stages, it is possible that HCGD requires a different learning rate schedule to achieve the same level of gradient noise. HCGD thus decreases the test error at a given learning rate, but needs to be trained at a higher learning rate to achieve the same level of gradient noise.

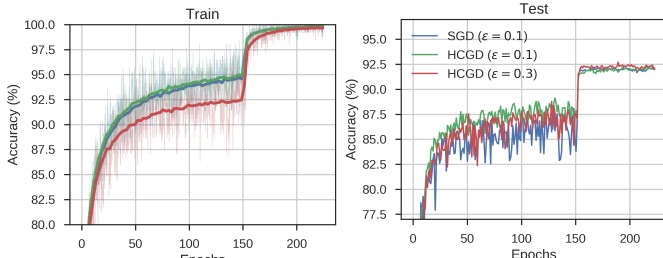

Figure 5: Results of a Squeezenet v1.1 trained on CIFAR10. The learning rate $\epsilon$ is decreased by a factor of 10 at epoch 150. For the train error we overlay the running average of each trace for clarity.

We next tested the performance of HCGD on a recurrent task. We trained an LSTM on the sequential MNIST task, in which pixels are input one at a time. The order of the pixels was permuted to further complicate the task. We found that HCGD outperformed SGD (Figure 6. We used 1 correction step, as before, but found that using more correction steps yielded even better performance. However, HCGD underperformed ADAM. While not the ideal optimizer for this task, the fact that SGD can be improved indicates that SGD does not move as locally in function space as it should. Parameter space thus a poor proxy for function space in recurrent networks.

HCGD first proposes an update by SGD, and then corrects it, but the first update step can also be other optimizers. Since Adam worked well for the sequential MNIST task, we tested if Adam could also be improved by taking a step to penalize the change in function space. We found that this is indeed the case, and show the results as well in Figure 6. To differentiate the SGD- and Adam-based methods, we refer to in the figure as SGD+HC and Adam+HC. This combination of Adam and $L^2$ functional regularization could help to achieve state-of-the-art performance on recurrent tasks.

## 4 DISCUSSION

Neural networks encode functions, and it is important that analyses discuss the empirical relationship between function space and the more direct parameter space. Here, we argued that the $L^2$ Hilbert space defined over an input distribution is a tractable and useful space for analysis. We found that networks traverse this function space qualitatively differently than they do parameter space. Depending on the situation, a distance of parameters cannot be taken to represent a proportional distance between functions.

We proposed two possibilities for how the $L^2$ distance could be used directly in applications. The first addresses multitask learning. By remembering enough examples in a working memory to accurately

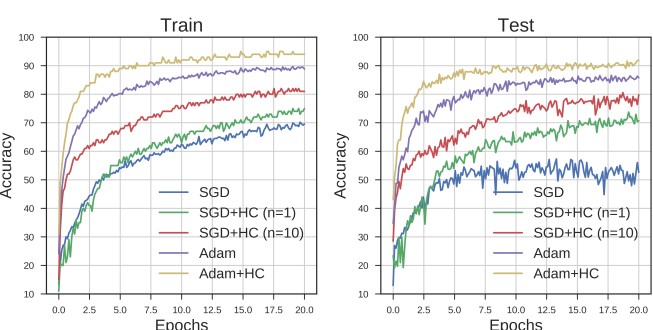

Figure 6: Results of a single-layer LSTM with 128 hidden units trained on the sequential MNIST task with permuted pixels. Shown are the traces for SGD and Adam (both with learning rate 0.01). We then take variants of the HCGD algorithm in which the first proposed step is taken to be an SGD step (SGD+HC) or an Adam step (Adam+HC). For SGD+HC we also show the effect of introducing more iterations $n$ in the SGD+HC step.

estimate an $L^2$ distance, we can ensure that the function (as defined on old tasks) does not change as a new task is learned. This regularization term is agnostic to the architecture or parameterization of the network. We found that this scheme outperforms simply retraining on the same number of stored examples. For large networks with millions of parameters, this approach may be more appealing than comparable methods like EWC and SI, which require storing large diagonal matrices.

We also proposed a learning rule that reduces movement in function space during single-task optimization. Hilbert-constrained gradient descent (HCGD) constrains the change in $L^2$ space between successive updates. This approach limits the movement of the encoded function in a similar way as gradient descent limits movement of the parameters. It also carries a similar intuition as the forgetting application: to learn from current examples only in ways that will not affect what has already been learned from other examples. HCGD can increase test performance at image classification in recurrent situations, indicating both that the locality of function movement is important to SGD and that it can be improved upon. However, HCGD did not always improve results, indicating either that SGD is stable in those regimes or that other principles are more important to generalization. This is by no means the only possibility for using an $L^2$ norm to improve optimization. It may be possible, for example, to use the norm to regularize the confidence of the output function (e.g. Pereyra et al. (2017)). We are particularly interested in exploring if more implicit, architectural methods, like normalization layers, could be designed with the $L^2$ norm in mind.

It interesting to ask if there is support in neuroscience for learning rules that diminish the size of changes when that change would have a large effect on other tasks. One otherwise perplexing finding is that behavioral learning rates in motor tasks are dependent on the direction of an error but independent of the magnitude of that error (Fine & Thoroughman, 2006). This result is not expected by most models of gradient descent, but would be expected if the size of the change in the output distribution (i.e. behavior) were regulated to be constant. Regularization upon behavioral change (rather than synaptic change) would predict that neurons central to many actions, like neurons in motor pools of the spinal cord, would learn very slowly after early development, despite the fact that their gradient to the error on any one task (if indeed it is calculated) is likely to be quite large. Given our general resistance to overfitting during learning, and the great variety of roles of neurons, it is likely that some type of regularization of behavioral and perceptual change is at play.

## CODE AVAILABILITY

A Pytorch implementation of the HCGD optimizer can be found at https://github.com/KordingLab/hilbert-constrained-gradient-descent.

## ACKNOWLEDGMENTS

The authors would like to thank Roozbeh Farhoodi for helpful conversations, Mohammad Pezeshki for the suggestion to use the Adam optimizer to produce the proposed step within HCGD, and NIH grant number MH103910.

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

# A   CIFAR-10 $L^2$ and $\ell^2$ comparison

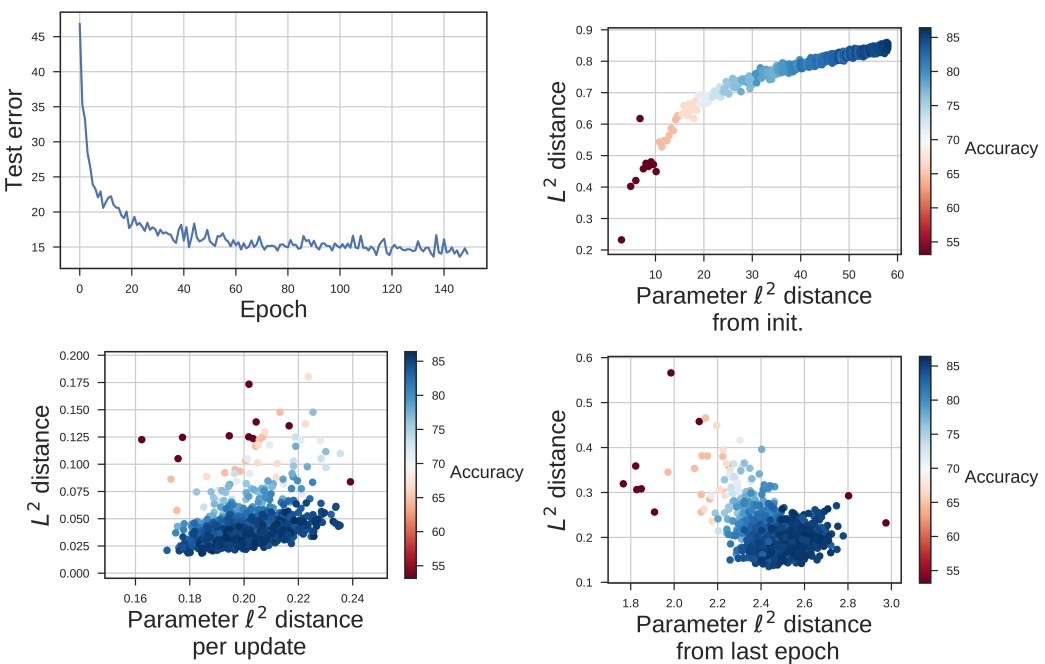

Figure A.1: This figure reproduces Figure 2, but includes the test error. The color scale is now also the test accuracy, rather than epoch number. Note that those epochs with qualitatively different $L^2/\ell^2$ ratios than the late optimization correspond to the epochs where test error is changing fastest.

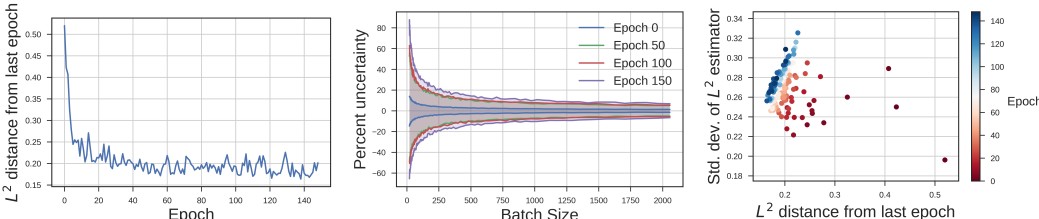

Figure A.2: This figure completes Figure 3 to include the standard deviation of the estimator for the distance between epochs. The scale of the standard deviation is similar to that of the $L^2$ estimator between batches, requiring near 1,000 examples for accuracies within a few percent.

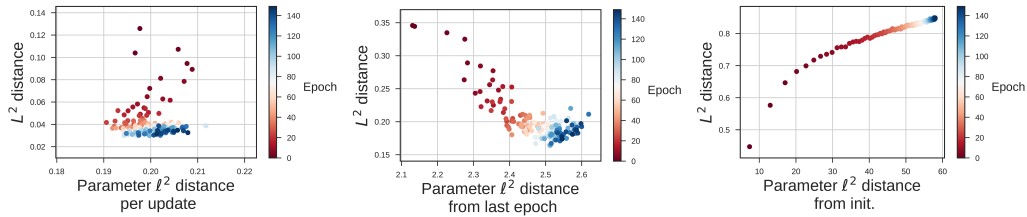

Figure A.3: Same as Figure 2 ($L^2/\ell^2$ ratio for three distance scales) but with all points within an epoch averaged. This makes the overall trends more apparent.

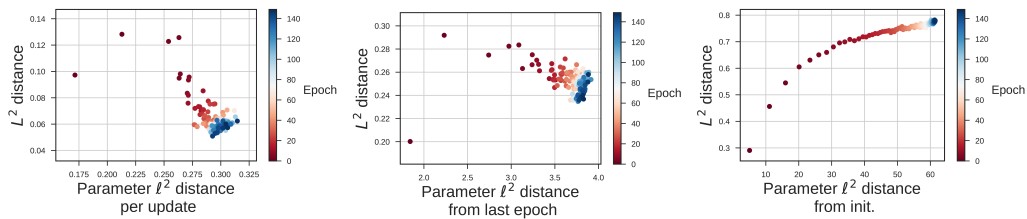

Figure A.4: Same as above, but for a network trained without Batch Normalization (BN). The change is most apparent in the x-axis scale of the left and middle plots. Without BN, larger parameter changes yield the same magnitude of $L^2$ changes, both between updates and between epochs. Furthermore, the $L^2/\ell^2$ ratio for the distance between updates (leftmost plot) changes less between epochs when BN is used. This appears largely a consequence of BN keeping the typical update size fixed at a more standard magnitude (and yet achieving a similar functional change.

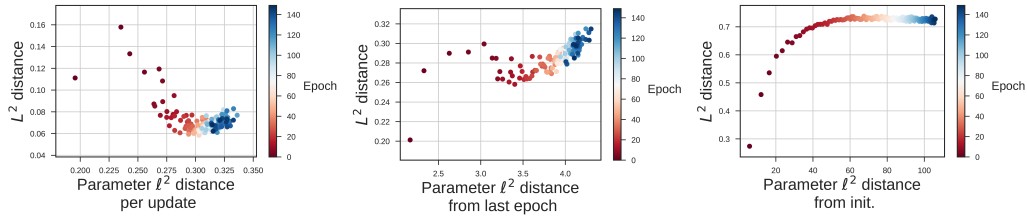

Figure A.5: Same as above, but for a network trained without Batch Normalization and also without weight decay. Weight decay has a strong effect. The main effect is that decreases the $\ell^2$ distance traveled at all three scales (from last update, last epoch, and initialization), especially at late optimization. This explains the left column, and some of the middle and right columns. (It is helpful to look at the "white point" on the color scale, which indicates the point halfway through training. Note that parameter distances continue to change after the white point when WD is not used). An additional and counterintuitive property is that the $L^2$ distance from the last epoch increases in scale during optimization when WD is not used, but decreases if it is. These comparisons show that WD has a strong effect on the $L^2/\ell^2$ ratio, but that this ratio still changes considerable throughout training. This is in line with this paper's motivation to consider $L^2$ distances directly.

# B COMPARING FUNCTION AND PARAMETER SPACES DURING MNIST OPTIMIZATION

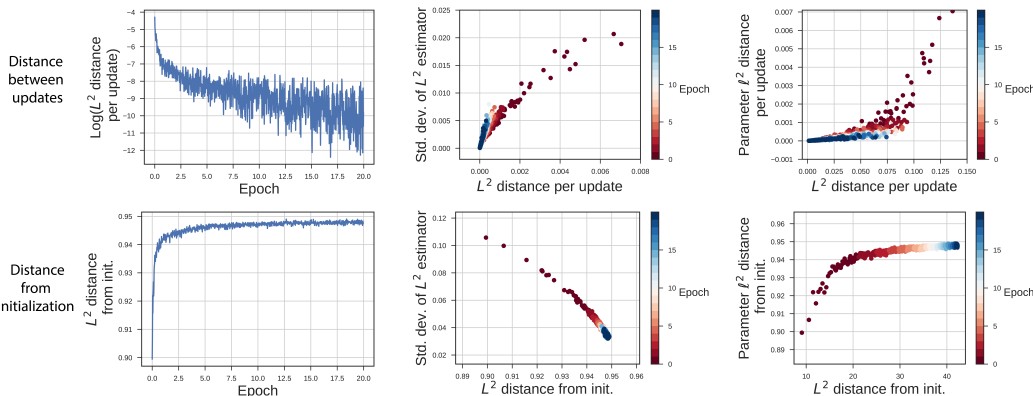

Figure B.6: Here we reproduce the results of Figure 2 and Figure 3 for the MNIST task, again using a CNN with batch normalization trained with SGD with momentum. It can be seen first that the majority of function space movement occurs very early in optimization, mostly within the first epoch. The standard deviation of the $L^2$ estimator, which sets the number of examples needed to accurately estimate a consistent value, is somewhat higher than for CIFAR-10. Finally, at right, it can be seen that the relationship between parameter distance traveled and function distance is similar to that of a CNN on CIFAR-10, include the qualitative change after test error converges (which here is around epoch 1).

## C  HCGD DECREASES THE DISTANCE TRAVELED IN $L^2$ SPACE

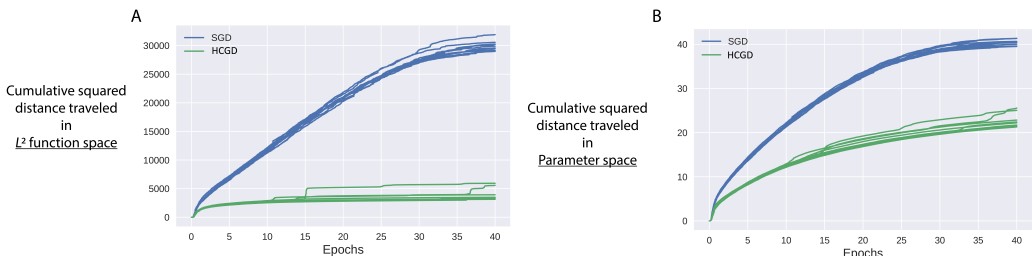

Figure C.7: The HCGD algorithm is designed to reduce motion through L2-space. To confirm this, here we plot the cumulative squared distance traveled during optimization for a simple MLP trained on MNIST. This is calculated by the simple cumulative sum of the squared distances between consecutive updates. (The squared distance is nice because Brownian motion will present as a linear increase in its cumulative sum). It can be seen that SGD continues to drift in L2-space during the overfitting regime (around epoch 15, which is when test error saturates), while HCGD plateaus. This indicates that the function has converged to a single location; it ceases to change. With SGD, on the other hand, the network continues to cahnge even long after test error saturates. It is interesting to note that HCGD allows the parameters to continue to drift even though the function has generally converged.

## D    DETAILED HCGD ALGORITHM

This version of the algorithm includes momentum. It also allows for multiple corrections.

---

Algorithm 2: Hilbert-constrained gradient descent. Implements Equation 7.

---

**Require:** $n \geq 1$          ▷ Number of corrective steps. May be 1.
**Require:** $\epsilon$          ▷ Overall learning rate
**Require:** $\eta$          ▷ Learning rate for corrective step
**Require:** $\beta$          ▷ Momentum

1: **procedure**
2:      $\theta \leftarrow \theta_0$          ▷ Initialize parameters
3:      $v \leftarrow 0$          ▷ Initialize momentum buffer
4:      **while** $\theta_t$ not converged **do**
5:          reset dropout mask, if using
6:          draw $X \sim \mathbb{P}_x$          ▷ Draw training batch
7:          $J \leftarrow \nabla_\theta C_0(X)$          ▷ Calculate gradients
8:          $v \leftarrow \beta v + \epsilon J$
9:          $\Delta\theta_0 \leftarrow -v$          ▷ Proposed update

10:          draw $X_V \sim \mathbb{P}_x$          ▷ Draw validation batch

11:          $g_{L^2} \leftarrow \nabla_{\Delta\theta}\big(\frac{\lambda^2}{N}\sum_{i=0}^{N}|f_{\theta_t}(x_i) - f_{\theta_t+\Delta\theta}(x_i)|^2\big)^{1/2}$

12:          $\Delta\theta_1 \leftarrow \Delta\theta_0 - \eta(g_{L^2})$          ▷ First correction
13:          $v \leftarrow v + \eta(g_{L^2})$          ▷ Update buffer
14:          **for** $1 < j < n$ **do**          ▷ Optional additional corrections

15:          $g_{L^2} \leftarrow J + \nabla_{\Delta\theta}\big(\frac{\lambda^2}{N}\sum_{i=0}^{N}|f_{\theta_t}(x_i) -$
16:          $f_{\theta_t+\Delta\theta_{j-1}}(x_i)|^2\big)^{1/2}$

17:          $\Delta\theta_j \leftarrow \Delta\theta_{j-1} - \eta(g_{L^2})$
18:          $v \leftarrow v + \eta(g_{L^2})$
19:          $\theta_t \leftarrow \theta_{t-1} + \Delta\theta$
20:      **return** $\theta_t$

---

## E    NATURAL GRADIENT BY GRADIENT DESCENT

In order to better compare the natural gradient to the Hilbert-constrained gradient, we propose a natural gradient algorithm of a similar style.

Previous work on the natural gradient has aimed to approximate $F^{-1}$ as best and as cheaply as possible. This is equivalent to minimizing Equation 2 (i.e. $J\Delta\theta + \frac{\lambda}{2}\Delta\theta^T F\Delta\theta$) with a single iteration of a second-order optimizer. For very large neural networks, however, it is much cheaper to calculate matrix-vector products than to approximately invert a large matrix. It is possible that the natural gradient may be more accessible via an inner gradient descent, which would be performed during each update step as an inner loop.

We describe this idea at high level in Algorithm 2. After an update step is proposed by a standard optimizer, the algorithm iteratively corrects this update step towards the natural gradient. To start with a good initial proposed update, it is better to use a fast diagonal approximation of the natural gradient (such as Adagrad or RMSprop) as the main optimizer. Each additional correction requires just one matrix-vector product after the gradients are calculated. Depending on the quality of the proposed update, the number of iterations required is likely to be small, and even a small number of iterations will improve the update.

---

Algorithm 3: Natural gradient by gradient descent. This algorithm can be paired with any optimizer to increase its similarity to the natural gradient.

---

**Require:** $n$         ▷ Number of corrective steps. May be 1.
**Require:** $\eta$         ▷ Learning rate for corrective step
  1: **procedure**
  2:      $\theta \leftarrow \theta_0$         ▷ Initialize parameters
  3:      **while** $\theta_t$ not converged **do**
  4:         $\Delta\theta_0 \leftarrow \text{RMSprop}(\theta_t)$         ▷ Use any optimizer to get proposed update
  5:         **for** $i < n$ **do**         ▷ Corrective loop
  6:            $\Delta\theta_{i+1} = \Delta\theta_i - \eta(J + \lambda F \Delta\theta_i)$         ▷ Step towards $\frac{1}{\lambda} F^{-1} J$
  7:         $\theta \leftarrow \theta + \Delta\theta$
  8:      **return** $\theta_t$

---

Since the Fisher matrix $F$ can be calculated from the covariance of gradients, it never needs to be fully stored. Instead, for an array of gradients $G$ of size (# parameters, # examples), we can write

$$F\Delta\theta = (GG^T)\Delta\theta = G(G^T \Delta\theta) \tag{7}$$

The choice of $G$ is an important one. It cannot be a vector of aggregated gradients (i.e. $J$), as that would destroy covariance structure and would result in a rank-1 Fisher matrix. Thus, we must calculate the gradients on a per-example basis. To compute $G$ efficiently it is required that a deep learning framework implement forward-mode differentiation, which is currently not supported in popular frameworks.

If we choose $G$ to be the array of per-example gradients on the minibatch, $F$ is known as the 'empirical Fisher'. As explained in Martens (2014) and in Pascanu & Bengio (2013), the proper method is to calculate G from the predictive (output) distribution of the network, $\mathbb{P}_\theta(y|x)$. This can be done as in Martens & Grosse (2015) by sampling randomly from the output distribution and re-running backpropagation on these fictitious targets, using (by necessity) the activations from the minibatch. Alternatively, as done in Pascanu & Bengio (2013), one may also use unlabeled or validation data to calculate $G$ on each batch.

