# OpenReview forum: "Measuring and regularizing networks in function space"
_ICLR.cc/2019/Conference_

### Official Review · AnonReviewer3 · 2018-11-01
**Core idea is interesting, but the follow-through is kind of scattered with weak results in too many directions.**

**Rating:** 6
**Confidence:** 4

**Review:**

This paper proposes a method for functional regularization for training neural nets, such that the sequence of neural nets during training is stable in function space. Specifically, the authors define a L2 norm (i.e., a Hilbert norm), which can be used to measure distances in this space between two functions. The authors argue that this can aid in preventing catastrophic forgetting, which is demonstrated in a synthetic multi-task variant of MNIST.   The authors also show how to regularize the gradient updates to be conservative in function space in standard stochastic gradient style learning, but with rather inconclusive empirical results.  The authors also draw upon a connection to the natural gradient.


***Clarity***

The paper is reasonably well written.  I think the logical flow could be improved at places.   I think the major issue with clarity is the title.  The authors use the term "regularizing" in a fairly narrow sense, in particular regularizing the training trajectory to be stable in function space.  However, the more dominant usage for regularizing is to regularize the final learned function to some prior, which is not studied or even really discussed in the paper.

Detailed comments:

-- The notation in Section 2 could be cleaned up.  The use of \mu is a bit disconnected from the rest of the notation.

-- Computing the empirical L2 distance accurately can also be NP hard.  There's no stated guarantee of how large N needs to be to have a good empirical estimate.  Figure 3 is nice, but I think a more thorough discussion on this point could be useful.

-- L2-Space was never formally defined.

-- Section 2.1 isn't explained clearly.  For instance, in the last paragraph, the first sentence states "the networks are initialized at very different point", and halfway into the paragraph a sentence states "all three initializations begin at approximately the same point in function space.".  The upshot is that Figure 1 doesn't crisply capture the intuition the authors aim to convey.


***Originality***

Strictly speaking, the proposed formulation is novel as far as I am aware.  However, the basic idea has been the air for a while.  For instance, there are some related work in RL/IL on functional regularization:
-- https://arxiv.org/abs/1606.00968

The proposed formulation is, in some sense, the obvious thing to try (which is a good thing).  The detailed connection to the natural gradient is nice.  I do wish that the authors made stronger use of properties of a Hilbert space, as the usage of Hilbert spaces is fairly superficial.  For instance, one can apply operators in a Hilbert space, or utilize an inner product.  It just feels like there was a lost opportunity to really explore the implications.


***Significance***

This is the place where the contributions of this paper are most questionable.  While the multi-task MNIST experiments are nice in demonstrating resilience against catastrophic forgetting, the experiments are pretty synthetic.  What about a more "real" multi-task learning problem?

More broadly, it feels like this paper is suffering from a bit of an identity crisis.  It uses regularizing in a narrow sense to generate conservative updates.  It argues that this can help in catastrophic forgetting.  It also shows how to employ this to construct the standard bounded-update gradient descent rules, although without much rigorous discussion for the implications.  There are some nice empirical results on a synthetic multi-task learning task, and inconclusive results otherwise.  There's a nice little discussion on the connection to the natural gradient.  It argues that that this form of regularization lives in a Hilbert space, but the usage of a Hilbert space is fairly superficial.  All in all, there are some nice pieces of work here and there, but it's all together neither here or there in terms of an overall contribution.


***Overall Quality***

I think if the authors really pushed one of the angles to a more meaningful contribution, this paper would've been much stronger.  As it stands, the paper just feels too scattered in its focus, without a truly compelling result, either theoretically or empirically.

---

> ### Author Response · Authors · 2018-11-19
> **We have made some edits to address these concerns**
>
> We are grateful for the close reading and helpful review.
>
> It is true that we pursued many directions in this work, and that these results somewhat compete for space. Our high-level goal was to establish (and disseminate) that 1) yes, there are other measures of function space besides the Fisher metric, 2) the L2 distance is actually reasonable to estimate, and 3) this could have many direct applications. We could have more aggressively documented each application, but we thought this would distract from the overall message.
>
> With that said, we present three specific empirical results that we believe are quite significant.
>
> 1)    Near state-of-the-art results on catastrophic forgetting at considerable less computational cost.
>
>
> To this point, there is a significant advantage of our method that we did not previously note. Unlike the benchmark methods of SI and EWC, our method does not require knowledge of task boundaries. SI and EWC both require this input, but in many real-world applications this is not available, or tasks shift continuously. We have now noted this in the text.
>
> The reviewer noted that our task is somewhat synthetic. While this is true, it was the same task as was presented in the papers of both of our benchmark methods.
>
>
> 2)    Better performance than ADAM at training recurrent neural networks. This is a new result.
>
> Previously, the HCGD algorithm adjusted the updates proposed by SGD, and we showed that on the sequential MNIST task that this adjustment outperformed SGD. We have introduced a new variant of the HCGD algorithm that adjusts the updates proposed instead by ADAM. We document that this scheme improves upon ADAM on the sequential MNIST task.
>
> 3)    The empirical study relating L2 distances to l2 distances. It is surprising to us that this has not been done before, given how often methods are designed to operate on parameters. These figures communicate a compelling finding that could be quickly digested but that could nevertheless change how researchers would design a new learning rule, or trust theoretical results relying on Lipschitz bounds.
>
> We believe that these findings are significant, and that they belong together.
>
> Responses to individual comments:
>
> ‘I think the major issue with clarity is the title.  The authors use the term "regularizing" in a fairly narrow sense, in particular regularizing the training trajectory to be stable in function space.  However, the more dominant usage for regularizing is to regularize the final learned function to some prior, which is not studied or even really discussed in the paper.”
>
> It is true that our main algorithm is more correctly a form of trajectory constraint that than imposing a prior. For this reason our algorithm is called “Hilbert-constrained gradient descent” rather than, say, “Hilbert-regularized”. Still, there were a few points that we used “regularization” in this broad sense, and we have gone through and edited these for language. In the title, however, we played around with several options and thought that among them this was the best balance of precision and clarity.
>
>
> “The use of \mu is a bit disconnected from the rest of the notation.”
> We don’t observe any typos in this section, and this is standard notation to denote a probability measure. We are not sure we understand this comment, though, and welcome any clarification.
>
>  “Computing the empirical L2 distance accurately can also be NP hard. There's no stated guarantee of how large N needs to be to have a good empirical estimate. Figure 3 is nice, but I think a more thorough discussion on this point could be useful.”
>
> This is true; we realize now that the text implied that our the convergence scales less than exponentially with N when this might not be the case. Unfortunately, precise guarantees of convergence will depend both on the data distribution and also upon the network. This is why we took an empirical approach. We have removed the implied claim that arbitrarily precise estimates are not NP-hard.
>
> “L2-Space was never formally defined.”
> At the moment we define the space at the very start of Section 2 by writing the norm that defines the space. It is true that Hilbert spaces are defined by the inner product, but since this is a standard function space we thought the inner product would be apparent from the norm. We have updated the manuscript such that we mention the inner product, as well.
>
> “Section 2.1 isn't explained clearly. For instance, in the last paragraph, the first sentence states "the networks are initialized at very different point", and halfway into the paragraph a sentence states "all three initializations begin at approximately the same point in function space.". The upshot is that Figure 1 doesn't crisply capture the intuition the authors aim to convey”
>
> We meant different points in parameter space, which correspond to a similar point in function space. We have edited this paragraph to be more clear about its overall relevance.

---

> > ### Comment · AnonReviewer3 · 2018-11-22
> > **Thanks, some more comments**
> >
> > 1) Ok, I withdraw the synthetic experiments as a negative point against the paper.
> >
> > 2) I'm not so sure about this one.  While I accept the synthetic catastrophic forgetting experiment as an established (although perhaps quickly expiring) setup, I'm not sure about the RNN setting.  To make a broad claim ("Better performance than ADAM at training recurrent neural networks. This is a new result."), one needs to establish it on a broad set of setups (NLP tasks, time series, etc.).  Otherwise, I don't view this as a significant result, but simply a curious result that might be significant.
> >
> > 3) Yes, I do agree with this point.  And upon reflection, this result is more significant than I previously realized.
> >
> > "In the title, however, we played around with several options and thought that among them this was the best balance of precision and clarity."
> > I actually think the current title overly maximizes recall at the expense of precision, and thus is not clear.  The title implies a broad claim on function regularization which is not reflected in the paper.
> >
> > Conventional gradient descent is actually a special case of Mirror Descent with a Euclidean proximity term.  See:
> > -- http://www.princeton.edu/~yc5/ele538_optimization/lectures/mirror_descent.pdf
> > -- https://static.googleusercontent.com/media/research.google.com/en//pubs/archive/37013.pdf
> > What this submission is essentially proposing is a proximity term in function space.  The authors decide to invoke the term Hilbert space, and in doing so I was expecting some deeper treatment of the topic.
> >
> > Regarding \mu:  One can remove the first two equations in Section 2 and not lose anything in the exposition.  Just start with the third equation.  The paper doesn't use the measure-theoretic notation \mu in any interesting way, it just adds more notation.  \mu appears nowhere else in the paper.
> >
> >
> > In summary, I am willing to raise my rating to 6 (weak accept).

---

### Official Review · AnonReviewer1 · 2018-11-02
**Nice empirical motivations but weak proposed solution**

**Rating:** 6
**Confidence:** 4

**Review:**

Summary:
This paper proposes first to measure distances, in a L2 space, between functions computed by neural networks. It then compares those distances with the parameter l2 distances of those networks, and empirically shows that the l2 parameter distance is a poor proxy for distances in the function space. Following those observations, the authors propose to use such constraint to combat catastrophic forgetting, and show some results on the permuted MNIST task. Finally, they propose the Hillber-constrained gradient descent (HCGD), a gradient descent algorithm that constraint movement in the function space, and evaluate it on a CNN (CIFAR10) and an LSTM (permuted MNIST).

Clarity:
The paper is well motivated, clearly written and easy to follow.

Novelty:
The idea of trying to move in the function space rather than in the parameter space is definitely not new (see the whole literature about Natural Gradient for instance). However, the proposed HCGD seems quite new, but unfortunately it doesn’t seem to perform well.

Pros and Cons:
+ The paper is well motivated, not only through the text but also with empirical evidence (section 2).
+ The paper focuses on an important research direction in deep learning.
+ This paper proposes a novel algorithm that penalizes movement in the function space.
- However, it is not clear if the proposed algorithm actually penalizes the distance in function space, since it is performing a crude approximation of the distance measure (using one step of gradient).
- Better way of penalizing movement in the function space already exists (at least for probability distributions: Natural Gradient)

Detailed Comments:
1. Batch Normalization and Weight Decay:
I have mixed feelings about your experiments in section 2. Both Batch Normalization (BN) and Weight Decay (WD) have a regularization effect on the weights.  I am wondering if the change in ratio L2/l2 during the course of training is simply caused by the regularization terms getting stronger and stronger (compared to the cross-entropy loss). Also, BN makes the function computed by the network independent of the scale (of each row) of the weight matrices. I do think that running again those experiments without BN and WD would make the argument that “the parameter space is a proxy for function space” more robust.
2. About HCGD:
The origins of the HCGD algorithm is extremely similar to the origins of Natural Gradient (NG) (just switch the L2 norm with the KL). The main difference resides in how the proximal formulation (equation 2) is approximated. For NG, one approximate the KL using a 2nd order Taylor expansion and then the proximal formulation is explicitly solved for Delta theta, where HCGD takes only a simple gradient step. It is thus not clear how well this step is  indeed a good approximation of the distance in function space. For CNNs and LSTMs, K-FAC [1-2], which is a Natural Gradient approximation, has been shown to outperform ADAM, so the proposed approximation might not be good enough, as HCGD doesn't beat ADAM in the experimental setup. One experiment that would be nice to have is to do one update of the parameter in a neural network (using HCGD) and then measure how much you actually moved in the function space.
[1] Roger Grosse, James Martens, A Kronecker-factored Approximate Fisher Matrix for Convolution Layers, ICML 2016
[2] James Martens, Jimmy Ba, Matt Johnson,Kronecker-factored Curvature Approximations for Recurrent Neural Networks, ICLR 2018

Minor Comments:
Section 2.3: “one would require require” -> “one would require”
Figure 3: “that a set batch size” -> “that a fixed batch size”
Section 3.1.1: “permuted different on” -> “permuted differently on”
Section 3.2.1: “that minimizes equation 6” -> “that minimizes equation 5”

Conclusion:
The paper proposes nice empirical evidence than parameter distance is not a good proxy for function distance. However, it is not clear if the proposed algorithm actually fixes this problem.

---

> ### Author Response · Authors · 2018-11-19
> **We have added new analyses on the L2/l1 ratio and tested whether the L2 distance is indeed decreased**
>
> We are grateful for the close reading and helpful review. We like your Pros and Cons list, and would like to respond to your two Cons.
>
> Re: “It is not clear if the proposed algorithm actually penalizes the distance...”:
>
> To address this, we have added a figure in Appendix C that shows that the algorithm decreases movement in function space, even for a single correction step.
>
> However, we think there is a small misunderstanding here. The number of correction steps taken doesn’t actually impact the quality of the approximation of the distance measure, but rather the quality of the constraint that the change in function space is constant for each step. The quality of the distance measure approximation is controlled only by the number of samples the measure is computed over, and we took care in section 2 to determine how many samples would be necessary.
>
> The extended algorithm in the Appendix current does have an option for multiple steps. We did not focus on this option because of the computational expense, and since a single step already shows some improvement. However, the algorithm does work better for larger number of correction steps. We have updated the sequential MNIST figure to make this clear.
>
> Re: “Better way of penalizing movement in the function space already exists (at least for probability distributions: Natural Gradient)":
>
> The natural gradient has its strengths, but we disagree that it is universally better. Our method has a few strengths over the natural gradient. First, it can be generalized to regularize the change in function space between any two arbitrary functions, while the natural gradient is set to regularize with respect to only local changes between updates. This is because the 2nd order Taylor expansion of the KL divergence is only valid locally. We exploited this advantage in our catastrophic forgetting section, and used the distance to regularize the functional change between tasks.
>
> The KL divergence also has different properties than the L2 norm, and is not the better choice in all circumstances. If the distributions of two networks are nonoverlapping, the KL divergence is infinite. Imagine, for example, that each output distribution is zero everywhere but a single line, and that the lines of the two distributions are parallel but separated. In this case the L2 norm is well-defined and gets smaller if you pull the lines closer to one another. The KL divergence is simply infinite until the lines overlap, at which point it becomes 0. This behavior is not likely to emerge in the natural gradient setting when the two networks have necessarily very close distributions, but in other settings (like the forgetting task, or otherwise when comparing far distributions like between that of a GAN’s output and real images) the L2 norm will be better.
>
> Detailed comments:
> 1.     We have run new experiments to examine how BN and WD affect the L2/l2 ratio. These now appear in Appendix A.3-5. As predicted, BN and WD both have strong effects on how the L2/l2 ratio changes throughout learning. However, their omission seems to actually exacerbate the problem, and the L2/l2 ratio still changes considerably. We discuss the changes in figure captions, and point out in the main text that we have run these controls.
>
> 2.
> Regarding the approximation quality:
> a.     The quality of the distance measure approximation depends on the number of validation examples of its empirical estimator, rather than the number of gradient steps. Even as formulated it can be arbitrarily accurate if one uses many examples.
> b.     The new Appendix figure measures this to make sure that the distance is indeed decreased.
>
> Regarding the comparison between HCGD and the natural gradient:
> a.     We have updated the sequential-MNIST task with a version of HCGD that bootstraps ADAM, rather than SGD. (Rather than taking a L2-regularizing step after an SGD step, we now take it after an ADAM step). Just as the first version outperformed SGD, this new version outperforms ADAM.
> b..     It often takes several refinements on a method before records are set. The natural gradient has been known for decades, and it is only recently with the additional modification of Kronecker factorization that it could be applied to large networks. Since ours is a fundamentally different approach to thinking about function space than the natural gradient, we actually consider this first attempt quite promising. We feel that it is important to get this work out so that a broader community can help think of potential improvements and modifications. Thus, we ask that this work be considered more as the initial introduction of a different approach, rather than a paper fine-tuning an established optimizer.
>
> Minor comments:
> Thank you for pointing out these errors. We have addressed them in the draft.

---

> > ### Comment · AnonReviewer1 · 2018-12-09
> > **Answer to your rebuttal**
> >
> > Thank you for your comments and the plots you added.
> >
> >
> > Re Re: “It is not clear if the proposed algorithm actually penalizes the distance...”:
> >
> > Thanks for the Figure C.7. Now we can see better how the two algorithms allow the networks to move in the function space, and that indeed HCGD penalizes movement in function space. 2 small comments: What are the different lines plotted (there are several green (and blue) lines)? Typo: cahnge -> change
> >
> > Also thank you for your precision about the quality of the approximation.
> >
> >
> > Re Re: “Better way of penalizing movement in the function space already exists (at least for probability distributions: Natural Gradient)":
> >
> > I agree with your first point that you can measure distance between any two functions and that you don’t suffer from the local validity of Natural Gradient. I do think however it is only valid in the case of catastrophic forgetting, not in the case of neural network training, since we need to make relatively small steps anyways, and thus we don’t break the locality assumption.
> >
> > About the KL divergence and Natural Gradient: I agree Natural Gradient might not be ideal in all cases (and doesn’t work if the network does not output distributions). But one could also, for instance, replace the Fisher by the Gauss-Newton approximation of the Hessian to approximate the distance measure between the two functions.
> >
> >
> > With that said, I am willing to raise my evaluation to a 6. Like R3, I find the idea novel and really interesting, but the paper should be refocused a bit. I understand the goal is to show that this method could be applied to a wide range of setups, but then it fails to be very convincing on those setups.

---

### Official Review · AnonReviewer2 · 2018-11-04
**Experimental results are not convincing**

**Rating:** 6
**Confidence:** 3

**Review:**

Although, I liked the exploratory part of the paper I must admit that I found myself confused a few times. The results given in the paper suggest that the proposed HCGD does not demonstrate any advantages on CIFAR-10 and has a limited impact on seq. MNIST. I think that section 3.3 of the paper should be extended and demonstrate some more convincing results.
Overall, I am not certain about my assessment. Therefore, I set my confidence level to "2: The reviewer is willing to defend the evaluation, but it is quite likely that the reviewer did not understand central parts of the paper".

Update on 17 Nov:

Section 2.
I am not sure that the results shown in Figure 2 tell more answers than they pose new questions.
For instance, "In particular, the parameter distance between successive epochs is negatively correlated with the L^2 distance for most of optimization (Fig. 2b). The distance from initialization shows a clean and positive relationship, but the relationship changes during optimization"
Would it be possible to have a supplementary figure with weight decay switched-off? I am not sure why you need it at all since the purpose is not to get state-of-the-art results. Could it also explain the angle for L^2/l^2 shown in the third column since weight decay is something that affects l^2?
I am not sure that the discussion of the negative correlation is sufficient. The actual correlation is linked to the stage of convergence, it would be nice to have a figure showing its average value per epoch (you say it is negative for the most part of optimization) and some discussion on its impact for the remaining part of your paper.

Section 3.
I am not an expert in online learning, this is probably why I don't recognize the novelty of the proposed approach. Is it novel to train networks for new tasks while making the objective function accounting for the old tasks? It sounds like a definition of online learning of multiple tasks. Importantly, here it is done while keeping training data from the old tasks. I understand your arguments about storage, but I find it surprising that your proposed change to the objective function is novel. If it is the case, please emphasize it more and mention that despite its simplicity, this idea is very novel. Otherwise, please cite relevant papers where similar methods were used.

I am not sure it is optimal to put Algorithm 1 in experimental results and applications. I don't see it as an application of your observations. I can imagine that the algorithm was inspired by your observations but it is your primary contribution and if possible should be discussed in a separate section. Here, you present it and then discuss how it is related to the natural gradient.
Please consider an alternative presentation where you first discuss the natural gradient and its various related works and algorithms, then present your algorithm and then demonstrate your empirical observations. This presentation might contradict the timeline of the development of your approach but it might help to better connect your work to other works  on the same topic. Also, it might help to better show novelties of your approach/observations.

Please comment if you find some interesting connection with [1].

[1] "Regularizing neural networks by penalizing confident output distributions" https://arxiv.org/pdf/1701.06548.pdf

Update on Nov 30:
I updated my score to 6 and my confidence level to 3.

---

> ### Author Response · Authors · 2018-11-19
> **New experimental results and general responses**
>
> We are grateful for the close reading and helpful review. We have made several changes to the paper in response.
>
> First, we have introduced a variant of the proposed algorithm that uses the Adam optimizer to take a proposed step, rather than a SGD step. We found that this outperforms standard Adam in training recurrent networks. In the sequential MNIST task, we had previously augmented only SGD with the L2 regularization, and saw that it boosted performance.
>
> Regarding section 3: The method is indeed simple, but has no exact precedence in the literature that we are aware of, either. We have updated the manuscript to underline that this is a novel approach. It performs well, too; our results are very near the state-of-the-art method of Synaptic Intelligence, despite being significantly cheaper and less memory-intensive. (To even test SI on our 64GB box, in fact, we had to decrease the size of the network). There is also one significant advantage of our method that we did not at all emphasize in the paper: it does not require knowing “task boundaries”, the moment when one task ends and another begins. Such knowledge is unavailable in many continual learning applications, including when tasks smoothly deform into one another instead of having sharp breaks. SI and EWC, the benchmark methods we compare to, both require this knowledge. We now emphasize this additional advantage in the paper.
>
> As for section 2, we want to first emphasize that we designed this section to be of wider interest than just to motivate our later algorithm. We received feedback that this work would be relevant to theoreticians whose work depends on the relationship between parameters and output functions. This is a common situation; parameters are easy to analyze and change predictably, while the output function determines performance. Section 2 is meant to appeal to the community of neural network researchers interested in empirical characterizations. Aside from our metric, we are not aware of other feasible methods to calculate the distance between two networks’ functions that works globally. (Only local measures, like the Fisher metric, exist). This is why we were initially more interested in establishing that our measure of distance in function space is actually feasible to calculate, and why we devoted such space to evaluating the convergence properties of its empirical estimator. It is for these researchers that we analyzed the relationship between these two distances, rather than just for exploratory purposes setting up Section 3.
>
> As you suggested, we investigated whether Figure 2 would change if the network were trained without weight decay. This now appears in Figure A.5. As you predicted, weight decay affects the angle L2/l2 in the third column; the no-WD network traverses larger distances in l2 space, and actually moves less in L2 space than the WD network. Weight decay affects the other columns, as well, and actually removes the negative correlation in the middle column. The L2/l2 ratio still changes considerable throughout training, which is in line with this paper's motivation to consider L2 distances directly. We also followed your other suggestion to present the figure when each epoch is averaged. This now appears as Figure A.3.
>
> Thank you for the suggestion to discuss the natural gradient first, as motivation. An early draft of this work did indeed frame the work like this. However, we later realized from feedback that the empirical characterizations of section 2 were of wider interest than because of their relation to the natural gradient literature. Furthermore, there are many uses of the L2 distance besides a natural-gradient-esque algorithm (such as mitigating catastrophic forgetting). Since much of the paper is not directly inspired by or meant to replace the natural gradient, we have decided not to lead with that concept.
>
> Lastly, thank you for the related reference. This paper penalizes the entropy of a network’s output distribution to reduce overconfident probabilities. It’s an interesting idea, and we cite it in the discussion.

---

### Public Comment · ~Frederik_Benzing1 · 2020-01-14
**Related Continual Learning Literature**

Thanks for providing this very interesting perspective on regularisation in both the context of continual learning and optimisation, I very much enjoyed the read and hope to come back to it in the future.

You point out that you are not aware of direct precedence in the [continual learning] literature for your algorithm - but I think there might be some.
If I understand your approach correctly (the approach given the last displayed equation on p.5), it results in continual learning algorithm which is very similar to `Learning without Forgettig' (https://arxiv.org/abs/1606.09282, 2016). Again, if i don't misunderstand something, the only difference between the approaches seems to be which data is used to regularise the net (data from new task for your approach and data from old task for LwF).
LwF in turn could be interpreted as a form knowledge distillation (https://arxiv.org/abs/1503.02531).



On a slightly different note,  your baseline experiment 'Adam+retrain' seems to be very relevant from a continual learning viewpoint. In [https://arxiv.org/pdf/1902.10486.pdf] it is suggested that this baseline can be very strong if set up carefully. Their experiments indicate that storing 15 examples per class (which roughly corresponds to the size of your cache), this approach outperforms both SI and EWC by a large margin. While your setup is slightly different from theirs, it might interesting to see how this version of Adam+retrain performs.

---

### Meta-Review · Area_Chair1 · 2018-12-18
**Borderline accept**

**Confidence:** 4
**Recommendation:** Accept (Poster)

**Metareview:**

This paper proposes to regularize neural network in function space rather than in parameter space, a proposal which makes sense and is also different than the natural gradient approach.

After discussion and considering the rebuttal, all reviewers argue for acceptance. The AC does agree that this direction of research is an important one for deep learning, and while the paper could benefit from revision and tightening the story (and stronger experiments); these do not preclude publishing in its current state.

Side comment: the visualization of neural networks in function space was done profusely when the effect of unsupervised pre-training on neural networks was investigated (among others). See e.g. Figure 7 in Erhan et al. AISTATS 2009 "The Difficulty of Training Deep Architectures and the Effect of Unsupervised Pre-Training". This literature should be cited (and it seems that tSNE might be a more appropriate visualization techniques for non-linear functions than MDS).